# Serum and Echocardiographic Markers May Synergistically Predict Adverse Cardiac Remodeling after ST-Segment Elevation Myocardial Infarction in Patients with Preserved Ejection Fraction

**DOI:** 10.3390/diagnostics10050301

**Published:** 2020-05-14

**Authors:** Tamara Pecherina, Anton Kutikhin, Vasily Kashtalap, Victoria Karetnikova, Olga Gruzdeva, Oksana Hryachkova, Olga Barbarash

**Affiliations:** Research Institute for Complex Issues of Cardiovascular Diseases, 6 Sosnovy Boulevard, 650002 Kemerovo, Russia; pechtb@kemcardio.ru (T.P.); kashvv@kemcardio.ru (V.K.); karevn@kemcardio.ru (V.K.); gruzov@kemcardio.ru (O.G.); hryaon@kemcardio.ru (O.H.); barbol@kemcardio.ru (O.B.)

**Keywords:** biomarkers, ST-segment elevation myocardial infarction, preserved left ventricular ejection fraction, reduced left ventricular ejection fraction, heart failure, NT-proBNP, soluble ST2, galectin-3, matrix metalloproteinases, tissue inhibitors of metalloproteinases

## Abstract

Improvement of risk scoring is particularly important for patients with preserved left ventricular ejection fraction (LVEF) who generally lack efficient monitoring of progressing heart failure. Here, we evaluated whether the combination of serum biomarkers and echocardiographic parameters may be useful to predict the remodeling-related outcomes in patients with ST-segment elevation myocardial infarction (STEMI) and preserved LVEF (HFpEF) as compared to those with reduced LVEF (HFrEF). Echocardiographic assessment and measurement of the serum levels of NT-proBNP, sST2, galectin-3, matrix metalloproteinases, and their inhibitors (MMP-1, MMP-2, MMP-3, TIMP-1) was performed at the time of admission (1st day) and on the 10th–12th day upon STEMI onset. We found a reduction in NT-proBNP, sST2, galectin-3, and TIMP-1 in both patient categories from hospital admission to the discharge, as well as numerous correlations between the indicated biomarkers and echocardiographic parameters, testifying to the ongoing ventricular remodeling. In patients with HFpEF, NT-proBNP, sST2, galectin-3, and MMP-3 correlated with the parameters reflecting the diastolic dysfunction, while in patients with HFrEF, these markers were mainly associated with LVEF and left ventricular end-systolic volume/diameter. Therefore, the combination of the mentioned serum biomarkers and echocardiographic parameters might be useful for the prediction of adverse cardiac remodeling in patients with HFpEF.

## 1. Introduction

Despite cardiovascular mortality trending to decline worldwide in the recent decade [1], it is still unacceptably high in Russian Federation [2,3]. Advances in the diagnosis and treatment of coronary artery disease (CAD), particularly ST-segment elevation myocardial infarction (STEMI), considerably reduced in-hospital but not long-term mortality rates, even in patients with preserved (≥50%) left ventricular ejection fraction (LVEF) [4]. The main cause of adverse outcomes upon MI is ventricular remodeling—a deposition of a disorganized extracellular matrix governed by activated resident fibroblasts—which ultimately results in chronic heart failure (CHF) [5,6]. Timely diagnostics of ventricular remodeling requires sensitive and specific biomarkers, which should be detectable in the serum, permitting noninvasive sample collection [7]. Unfortunately, albeit serum biomarkers are routinely measurable and can be combined with other diagnostic and prognostic modalities, they generally provide little information on the actual state of ventricular remodeling, which restricts their implementation into its management [8].

Promising serum markers of ventricular remodeling include N-terminal pro-B-type natriuretic peptide (NT-proBNP, a protein released from cardiomyocytes at myocardial stretch), soluble suppression of tumorigenicity 2 (sST2, a cytokine-related protein and an established marker of inflammation, hemodynamic stress, and cardiomyocytes strain), galectin-3 (an inflammatory β-galactoside-binding lectin acting as a matricellular protein), matrix metalloproteinases (MMPs) and their tissue inhibitors (TIMPs) regulating the profibrotic remodeling [7,8], and exosomes enriched by pro-fibrotic miRNA (miR-1, -21, -34a, -133, -192, -194, -208a, -425, -744) [9], although none of them have been recommended for the specific assessment of ventricular remodeling progression so far [8]. Hence, the search for novel systemic markers of ventricular remodeling is ongoing, and various patient cohorts are involved.

Currently, >70% of patients with CHF aged >65 years have a preserved LVEF (HFpEF), and this proportion tends to increase together with the rising prevalence of associated underlying health conditions (obesity, metabolic syndrome, and diabetes mellitus) [10]. In contrast to CHF with a reduced (<50%) LVEF (HFrEF), which progression can be directly assessed in a noninvasive manner by echocardiography, efficient monitoring of the cardiac status in the patients with HFpEF is challenging [11], although a number of studies in this direction have been reported [12,13]. Taken together, these facts underscore the importance of finding the informative and routinely measurable biomarkers of ventricular remodeling in this patient category [14]. Ideally, the candidate molecules should be involved in the development of ventricular remodeling rather than simply reflecting the stiffness of the heart extracellular matrix. In addition, a panel of serum biomarkers can be combined with echocardiographic parameters, possibly correlating with each other and thereby reinforcing the association with ventricular remodeling.

Here, we evaluated whether the combination of serum biomarkers (NT-proBNP, sST2, galectin-3, MMP-1, MMP-2, MMP-3, and TIMP-1) and echocardiographic parameters may be useful to predict the ventricular remodeling-related outcomes in patients with HFpEF as compared to those with HFrEF.

## 2. Materials and Methods

In this prospective study, we enrolled 100 consecutive patients with HFpEF and 154 patients with HFrEF who have been admitted to Research Institute for Complex Issues of Cardiovascular Diseases (Kemerovo, Russian Federation) in 2015. The investigation was carried out in accordance with the Good Clinical Practice and the Declaration of Helsinki. The study protocol was approved (date of approval: 04 February 2015) by the Local Ethical Committee of Research Institute for Complex Issues of Cardiovascular Diseases (Protocol No. 20150204). All patients provided a written informed consent after receiving a full explanation of the study. Criteria of inclusion were: (1) STEMI diagnosed by means of the respective European Society of Cardiology guidelines [14]; (2) age between 18 and 75; (3) successful percutaneous coronary intervention (PCI). Patients with no clinical signs of heart failure (Killip class I) and LVEF ≥ 40% were assigned to HFpEF group while those with heart failure clinical signs (Killip class to II-IV) and LVEF <40% were considered as having HFrEF. Criteria of exclusion were: (1) Diagnosed acute/chronic liver or kidney failure, cancer, chronic obstructive pulmonary disease, autoimmune, endocrine, or mental disorders; (2) STEMI occurred as a result of unsuccessful PCI or coronary artery bypass graft surgery; (3) in-hospital death of the patient. The study design is represented in Figure 1.

Thrombolytic therapy was applied in 11 (11%) patients. Biochemical profile and complete blood count were determined in all patients in addition to the collection of clinicopathological and demographic data from the case histories (Table 1).

Prior to MI, cardiovascular drugs were rarely used in patients with HFpEF (Table 2).

During the hospital stay, treatment was performed in accordance with the respective European Society of Cardiology guidelines [15]. Angina pectoris, chronic heart failure, atrial fibrillation, peripheral artery disease, arterial hypertension, diabetes mellitus/glucose intolerance, and hypercholesterolemia were diagnosed according to the respective ESC guidelines [16,17,18,19,20,21,22], while chronic kidney disease and overweight/obesity were defined as recommended in KDIGO [23] and NICE [24] guidelines. Past medical history of MI, stroke/transient ischemic attack, PCI or CABG surgery, smoking status, and family history of CAD were defined using the medical records. Coronary angiography was performed within the first hours after hospital admission using GE Healthcare Innova 3100 Cardiac Angiography System (General Electric Healthcare, Chicago, IL, USA). Although 26 out of 100 (26%) patients with HFpEF suffered from multivessel CAD and 17 out of 100 (17%) patients had complications during in-hospital stay, none of them died (Table 3).

Echocardiography (Sonos 2500 Diagnostic Ultrasound System, Hewlett Packard, Palo Alto, CA, USA) and enzyme-linked immunosorbent assay (ELISA) measurement of serum NT-proBNP (SK-1204, Biomedica, Vienna, Austria), sST2 (BC-1065X, Critical Diagnostics, San Diego, CA, USA), galectin-3 (BMS279/4, Thermo Fisher Scientific, Waltham, Massachusetts, MA, USA), matrix metalloproteinase (MMP)-1 (PDMP100, R&D Systems, Minneapolis, MN, USA), MMP-2 (PDMP200, R&D Systems, Minneapolis, MN, USA), MMP-3 (KAC1541, Thermo Fisher Scientific, Waltham, Massachusetts, MA, USA), and tissue inhibitor of metalloproteinases (TIMP-1, PDTM100, R&D Systems, Minneapolis, MN, USA) were performed on the 1st and 10th–12th day of hospitalization.

Statistical analysis was carried out utilizing Statistica 8.0 (Dell, Round Rock, TX, USA). Binary data (frequencies) were compared by Yates′s chi-squared test. A sampling distribution was assessed by the Shapiro-Wilk test. Descriptive data were represented by median and interquartile range (25th and 75th percentiles). Unpaired and serial (before-after) measurements were compared by Mann-Whitney U-test and Wilcoxon matched-pairs signed rank test, respectively. To assess the correlation, Spearman’s rank correlation coefficient was employed. *p* values ≤ 0.05 were regarded as statistically significant.

## 3. Results

As compared to patients with HFrEF, patients with HFpEF had significantly lower prevalence of past medical history of myocardial infarction, angina pectoris, atrial fibrillation, chronic heart failure, arterial hypertension, and chronic kidney disease, as well as major cardiovascular risk factors such as smoking, hypercholesterolemia, and family history of coronary artery disease (Table 1).

In patients with HFpEF, serial echocardiographic examination revealed a significant increase in LVEF, stroke volume, early mitral filling velocity to early diastolic mitral annular velocity ratio, and early mitral inflow velocity, along with a concurrent decrease in left ventricular end-diastolic volume, left ventricular end-systolic volume, left ventricular end-systolic diameter, deceleration time, left ventricular ejection time, early diastolic myocardial velocity, early to late diastolic myocardial velocity ratio, and early diastolic myocardial velocity to early mitral inflow velocity ratio from the 1st to the 10th–12th day after STEMI onset (Table 4), suggesting moderate improvement of the cardiac function.

In patients with HFpEF, serum levels of MMP-1, MMP-2, and MMP-3 did not exceed reference ranges at any of measured time points albeit augmented to the 10th–12th day after STEMI onset in comparison with the time of admission (Table 5). In contrast, TIMP-1 and galectin-3 were significantly higher at both of the time points, while NT-proBNP and sST2 demonstrated an increase exclusively on the first day of hospital stay (Table 5). Notably, serum concentrations of all elevated biomarkers (NT-proBNP, sST2, galectin-3, TIMP-1) were significantly reduced on the 10th–12th day after STEMI onset as compared with the time of admission (Table 5). The most pronounced decrease was documented for NT-proBNP and sST2 (3.8- and 1.8-fold, respectively).

Patients with HFrEF had normal levels of MMP-1, MMP-2, MMP-3, and TIMP-1 but elevated NT-proBNP, sST2, and galectin-3 at both of the time points, although the latter markers significantly reduced to the 10th–12th day after STEMI onset (Table 5). Notably, MMP-1, MMP-2, and MMP-3 increased to the 10th–12th day after STEMI onset, similar to the patients with HFpEF (Table 5).

In comparison with patients with HFrEF, those with HFpEF had significantly higher levels of NT-proBNP, sST2, galectin-3, MMP-2, and MMP-3 in conjunction with lower concentration of MMP-1 and TIMP-1 (Table 5).

Correlation analysis performed in regards to the patients with HFpEF (Figure 2) demonstrated that, at the time of admission, the serum concentration of sST2 directly correlated with a number of echocardiographic parameters (left ventricular end-diastolic (r = 0.42, *p* = 0.026) and end-systolic volume (r = 0.41, *p* = 0.030), as well as pulmonary artery pressure (r = 0.44, *p* = 0.001)). Likewise, serum galectin-3 on the first day after STEMI onset positively correlated with a myocardial performance (Tei) index (r = 0.41, *p* = 0.001). Serum NT-proBNP negatively correlated with early to late diastolic myocardial velocity ratio (r = −0.49, *p* = 0.001) on the 1st day and with LVEF on the 10th–12th day of hospital stay (r = −0.47, *p* = 0.007) and positively correlated with left ventricular end-systolic volume (r = 0.47, *p* = 0.002) left ventricular end-systolic diameter (r = 0.41, *p* = 0.020), and left ventricular end-diastolic pressure (r = 0.41, *p* = 0.030) at the latter time point.

At the time of admission, serum MMP-1 correlated with left ventricular mass to body surface area index (r = 0.46, *p* = 0.006), yet on the 10th–12th day after STEMI onset, TIMP-1, but not MMPs, showed a significant correlation (r = 0.56, *p* = 0.020). MMP-3, determined at the time of admission, negatively correlated with LVEF on the 10th–12th day of hospital stay (r = −0.68, *p* = 0.010), while MMP-1 positively correlated with left ventricular end-diastolic volume/body surface area (r = 0.57, *p* = 0.040) and left ventricular end-systolic volume/body surface area (r = 0.58, *p* = 0.030) if both were measured at the latter time point.

Patients with HFrEF generally had similar correlation trends (Figure 3). Serum NT-proBNP, sST2, galectin-3, and intriguingly, MMP-3, negatively correlated with LVEF at the time of admission (r = −0.57, *p* = 0.0019; r = −0.44, *p* = 0.0032; r = −0.45, *p* = 0.0021; r = −0.43, *p* = 0.0010, respectively), and similar trend was found regarding the hospital discharge. However, TIMP-1 directly correlated with LVEF (r = 0.49; *p* = 0.0003) and isovolumic relaxation time (r = 0.41; *p* = 0.0012) at the same time point.

Further, NT-proBNP and MMP-3 measured at the time of admission directly correlated with left ventricular end-systolic volume and diameter (r = 0.47, *p* = 0.0017 and r = 0.42, *p* = 0.0234, respectively, for NT-proBNP; r = 0.41, *p* = 0.0138 and r = 0.43, *p* = 0.0085, respectively, for MMP-3) at the 10th–12th day after STEMI onset. At the latter time point, MMP-3 also correlated with left ventricular end-systolic diameter (r = 0.42, *p* = 0.0026).

Therefore, serum markers well correlate with echocardiographic parameters at both of the studied time points in patients with both HFpEF and HFrEF are suggestive of a similar negative long-term prognosis in both of the groups despite better clinical symptoms in those with HFpEF in early HF stages. Hence, patients with HFpEF also have a high risk of adverse outcome.

## 4. Discussion

Unfortunately, preserved LVEF also does not guarantee the favorable outcome after STEMI and is frequently followed by CHF, arrhythmia, or heart block [25]. Multiple studies reported that patients with preserved LVEF after STEMI have similar prevalence of cardiovascular events as those with reduced LVEF [26], yet the data on available serum biomarkers informative of ventricular remodeling are still scarce [8]. Identification of patients with pathological cardiac remodeling and its sequelae is of crucial importance for the risk stratification.

Among the plethora of biomarkers, serum NT-proBNP is widely employed for the diagnostics of CHF, as well as control of treatment efficacy and prognostication in such patients [27]. Natriuretic peptides inhibit renin-angiotensin system and enhance sodium excretion, promoting peripheral vasodilation [28]. In our study, serum NT-proBNP was elevated at the time of admission and returned to the normal values to the 10th–12th day upon STEMI onset. In addition, we found a direct correlation of serum NT-proBNP with left ventricular end-systolic volume, left ventricular end-systolic diameter, and left ventricular end-diastolic pressure on 10th–12th day upon STEMI onset, along with the inverse correlation with early to late diastolic myocardial velocity ratio at the time of admission. Importantly, serum NT-proBNP also negatively correlated with LVEF on 10th–12th day upon STEMI onset. Previous studies well confirmed the role of serum NT-proBNP in predicting adverse cardiovascular outcome in patients with HFpEF [29,30].

The interleukin-1 receptor family member sST2 is considerably increased in response to the disrupted extracellular matrix homeostasis and subsequent cardiac remodeling [31,32]. In keeping with these data, serum sST2 declined from the 1^st^ to the 10th–12th day after STEMI onset and directly correlated with left ventricular end-diastolic and end-systolic volume along with pulmonary artery pressure at the time of admission. In other studies, serum sST2 was elevated in patients with congestive heart failure, correlating with left ventricular end-diastolic pressure and serum NT-proBNP [33]. Hypertensive patients with HFpEF had elevated serum sST2 which demonstrated better predictive value as compared with NT-proBNP (AUC 0.8 and 0.7, respectively), with 13.5 ng/mL as a cutoff in multivariate analysis [34]. Predicting efficacy of serum sST2 in relation to the outcomes was higher for HFpEF as compared to the patients with reduced LVEF, indicating a positive association with all-cause death and repeated HF-related hospitalization [35].

Galectin-3 is a beta-galactoside-binding pleiotropic lectin released into the extracellular matrix by activated cardiac macrophages and regulating apoptosis, proliferation, inflammation, and fibrosis [36,37], being notable for its association with left ventricular remodeling [38], incident heart failure [39], and its severity [37]. In patients with HFpEF, serum galectin-3 was significantly elevated as compared to the control subjects and strongly correlated with NT-proBNP [40,41]. Further, both serum galectin-3 and NT-proBNP well distinguished patients with HFpEF from healthy individuals in ROC analysis [41]. In contrast to NT-proBNP, serum galectin-3 retained its association with ventricular remodeling in HFpEF patients in multivariate analysis, although NT-proBNP was significantly associated with HF symptoms [42]. Regarding to the long-term outcomes, elevated galectin-3 at 6 or 12 months was associated with all-cause death and hospitalization independent of treatment arm and NT-proBNP [43]. We found an increase in serum galectin-3 at both time points, yet no significant correlations with echocardiographic parameters were detected excepting a positive correlation with myocardial performance (Tei) index on the first day after STEMI onset.

MMPs represent a large family of zinc-dependent proteases degrading extracellular matrix and are informative of ventricular remodeling following the first hours after STEMI onset [44]. However, it is unknown whether the initial elevation of serum MMPs and TIMP-1 affects further cardiac remodeling and development of CHF. Concentrations of MMP-9 and TIMP-1 are frequently augmented in patients with CHF, being associated with a number of echocardiographic parameters [45,46]. Notably, NT-proBNP and MMP-9 levels have been found increased at all time points from the acute phase until >4 years after STEMI, and their levels showed a significant correlation [47]. In addition, higher serum MMP-9 correlated with lower LVEF and functional myocardial mass in long-term survivors of complicated STEMI [47] and with increased left ventricular end-diastolic diameter and left ventricular wall thickness in Framingham study participants free of previous MI and CHF [48]. In relation to the HF progression, the predictive value of serum MMP-2 in patients with HFpEF was found to be higher as compared with NT-proBNP [49]. Here, we did not reveal a significant increase in MMPs in the early period upon STEMI, yet MMP-1 and TIMP-1 showed a positive correlation with left ventricular mass to body surface area index at sequential time points. Further, higher MMP-1 on the 10th–12th day after STEMI correlated with left ventricular end-diastolic volume and left ventricular end-systolic volume normalized to the body surface area while MMP-3 at the time of admission inversely correlated with LVEF before hospital discharge. Intriguingly, serum levels of MMP-2 and MMP-3, although being within the reference range, significantly augmented from the admission to the discharge regardless of the ejection fraction, suggesting these molecules are gradually released during ventricular remodeling. Further studies in this direction might address this point.

An important limitation of the study includes differences in risk factors and treatments between the patients resulting in a relatively heterogeneous variable. However, it was expected when considering the patient groups with distinct LVEF (HFpEF and HFrEF). A replication of our study on a larger sample would be beneficial for the field to confirm our results. Another limitation is relatively short observation interval (two weeks), which should possibly be extended in replicating studies.

Notably, patients with HFrEF had multiple correlations of serum markers (NT-proBNP, sST2, galectin-3, MMP-3) with LVEF, as well as left ventricular end-systolic volume and diameter, whereas in those with HFpEF, only NT-proBNP showed significant negative correlations with LVEF. Probably, patients with HFpEF share a number of pathophysiological pathways with those suffering from HFrEF, yet cardiac remodeling in the former patient category is related to the diastolic dysfunction rather than LVEF.

## 5. Conclusions

In the present study, we found a reduction in NT-proBNP, sST2, galectin-3, and TIMP-1 in patients with HFpEF and HFrEF from hospital admission to the discharge, as well as numerous correlations between the indicated biomarkers and echocardiographic parameters, testifying to the ongoing ventricular remodeling. In patients with HFpEF, studied biomarkers of ventricular remodeling (NT-proBNP, sST2, galectin-3, and MMP-3) correlated with the parameters, reflecting the diastolic dysfunction (myocardial stretch, transmitral flow profile, and myocardial performance (Tei) index). Contrariwise, in patients with HFrEF, these markers are mainly associated with LVEF and left ventricular end-systolic volume/diameter. Therefore, the combination of serum biomarkers and echocardiographic parameters might be useful for the prediction of adverse cardiac remodeling in patients with HFpEF, particularly after STEMI.

## Figures and Tables

**Figure 1 diagnostics-10-00301-f001:**
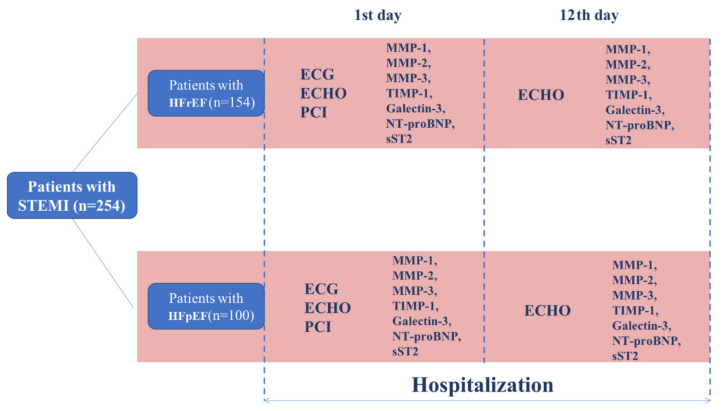
Study design. STEMI—ST-segment elevation myocardial infarction, HFpEF—STEMI with preserved ejection fraction, HFrEF—STEMI with reduced ejection fraction, ECG—electrocardiography, ECHO—echocardiography, PCI—percutaneous coronary intervention, MMP—matrix metalloproteinase, TIMP—tissue inhibitor of metalloproteinases, NT-proBNP—N-terminal pro-B-type natriuretic peptide, sST2—soluble suppression of tumorigenicity 2.

**Figure 2 diagnostics-10-00301-f002:**
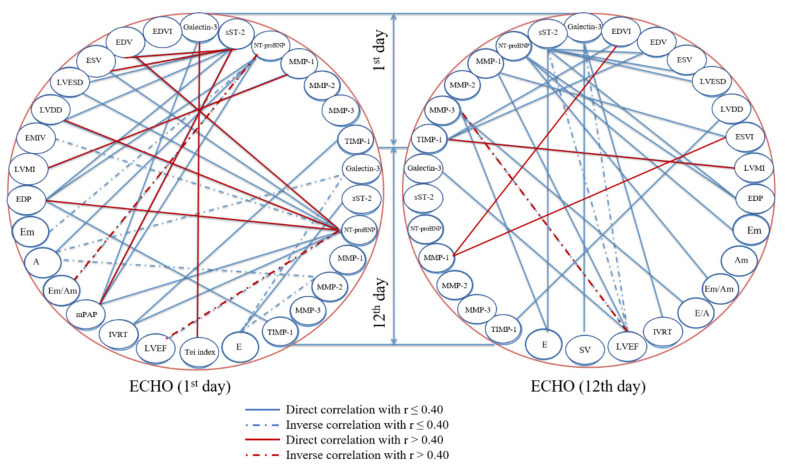
Correlation analysis of echocardiographic parameters and serum biomarkers in patients with ST-segment elevation myocardial infarction and preserved left ventricular ejection fraction (HFpEF, only statistically significant correlations with Spearman’s rho > 0.4–0.7 are marked red). Abbreviations used: A—late diastolic ventricular filling velocity, Am—late diastolic myocardial velocity, E—early diastolic ventricular filling, E/A—early to late diastolic ventricular filling velocity ratio, Em—early diastolic myocardial velocity, Em/Am—early to late diastolic myocardial velocity ratio, EDP—left ventricular end-diastolic pressure, EDV—left ventricular end-diastolic volume, EDVI—end-diastolic volume index (left ventricular end-diastolic volume/body surface area), EMIV—early mitral inflow velocity, ESV—left ventricular end-systolic volume (ESV), ESVI—end-systolic volume index (left ventricular end-systolic volume/body surface area), IVRT—isovolumic relaxation time, LVDD—left ventricular end-diastolic diameter, LVEF—left ventricular ejection fraction, LVESD—left ventricular end-systolic diameter, LVMI—left ventricular mass index (left ventricular mass/body surface area), mPAP—pulmonary artery pressure, MMP—matrix metalloproteinase, NT-proBNP—N-terminal pro-B-type natriuretic peptide, sST2—soluble suppression of tumorigenicity 2, SV—stroke volume, TIMP—tissue inhibitor of metalloproteinases.

**Figure 3 diagnostics-10-00301-f003:**
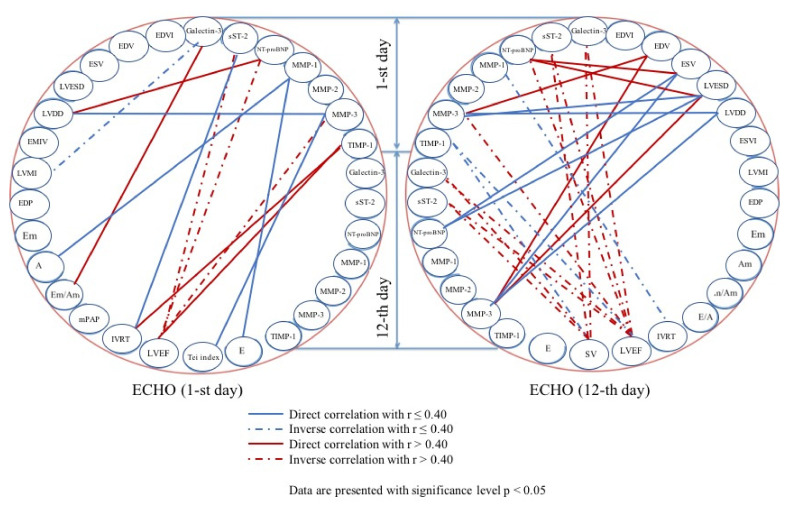
Correlation analysis of echocardiographic parameters and serum biomarkers in patients with ST-segment elevation myocardial infarction and reduced left ventricular ejection fraction (HFrEF, only statistically significant correlations with Spearman’s rho > 0.4–0.7 are marked red). Abbreviations used: A—late diastolic ventricular filling velocity, Am—late diastolic myocardial velocity, E—early diastolic ventricular filling, E/A—early to late diastolic ventricular filling velocity ratio, Em—early diastolic myocardial velocity, Em/Am—early to late diastolic myocardial velocity ratio, EDP—left ventricular end-diastolic pressure, EDV—left ventricular end-diastolic volume, EDVI—end-diastolic volume index (left ventricular end-diastolic volume/body surface area), EMIV—early mitral inflow velocity, ESV—left ventricular end-systolic volume (ESV), ESVI—end-systolic volume index (left ventricular end-systolic volume/body surface area), IVRT—isovolumic relaxation time, LVDD—left ventricular end-diastolic diameter, LVEF—left ventricular ejection fraction, LVESD—left ventricular end-systolic diameter, LVMI—left ventricular mass index (left ventricular mass/body surface area), mPAP—pulmonary artery pressure, MMP—matrix metalloproteinase, NT-proBNP—N-terminal pro-B-type natriuretic peptide, sST2—soluble suppression of tumorigenicity 2, SV—stroke volume, TIMP—tissue inhibitor of metalloproteinases.

**Table 1 diagnostics-10-00301-t001:** Clinicopathological features of the patients diagnosed with ST-segment elevation myocardial infarction with preserved ejection fraction (HFpEF, *n* = 100) and ST-segment elevation myocardial infarction with reduced ejection fraction (HFrEF, *n* = 154).

Features	Patients with HFpEF, *n* (%)	Patients with HFrEF, *n* (%)	*p* Value
Male gender	74 (74.0%)	102 (66.2%)	0.24
Past medical history of myocardial infarction	5 (5.0%)	70 (45.4%)	0.0001
Past medical history of percutaneous coronary intervention	3 (3.0%)	2 (1.3%)	0.62
Past medical history of coronary artery bypass graft surgery	0 (0.0%)	1 (0.6%)	1.0
Past medical history of angina pectoris	31 (31.0%)	108 (70.1%)	0.0001
Past medical history of atrial fibrillation	4 (4.0%)	28 (18.2%)	0.002
Past medical history of chronic heart failure	12 (12.0%)	101 (65.6%)	0.0001
Past medical history of stroke or transient ischemic attack	4 (4.0%)	18 (11.7%)	0.06
Peripheral artery disease	1 (1.0%)	10 (6.5%)	0.07
Arterial hypertension	70 (70.0%)	131 (85.1%)	0.006
Chronic kidney disease, stage 1–3	2 (2.0%)	15 (9.7%)	0.03
Type 2 diabetes mellitus	11 (11.0%)	30 (19.5%)	0.10
Glucose intolerance	2 (2.0%)	0 (0.0%)	0.15
Smoking	56 (56.0%)	57 (37.0%)	0.004
Hypercholesterolemia	22 (22.0%)	69 (44.8%)	0.0003
Overweight or obesity	71 (71.0%)	91 (59.1%)	0.07
Family history of coronary artery disease	3 (3.0%)	37 (24.0%)	<0.0001
**Median (Interquartile Range)**			
Age, years	57 (52; 63)	63 (56; 67)	<0.0001
Body mass index	26.9 (24.3; 29.8)	27.9 (25.1; 31.2)	0.0036
Duration of hospital stay	14 (12; 18)	15 (13; 19)	0.01

**Table 2 diagnostics-10-00301-t002:** The list of medications used in patients with ST-segment elevation myocardial infarction with preserved ejection fraction (HFpEF).

Drug	Prior to Myocardial Infarction Number of Patients (Equal to the Proportion as *n* = 100)	During the Hospital Stay Number of Patients (Equal to the Proportion as *n* = 100)
Aspirin	9	99
Clopidogrel	0	83
Ticagrelor	0	25
Beta-blockers	11	97
ACE inhibitors	11	77
Statins	4	94

ACE—angiotensin-converting enzyme.

**Table 3 diagnostics-10-00301-t003:** Coronary angiography features and in-hospital complications in patients with ST-segment elevation myocardial infarction with preserved ejection fraction (HFpEF).

Coronary Angiography Features	Number of Patients (Equal to the Proportion as *n* = 100)
>50% stenosis	41
Multivessel coronary artery disease (>3 affected arteries)	26
Complications during the hospital stay	
Arrhythmia or heart block	8
Pulmonary oedema	2
Radial artery bleeding	2
Coronary artery perforation	2
Early postoperative angina	1
Recurrent myocardial infarction	1
Cardiogenic shock	1
Death	0

**Table 4 diagnostics-10-00301-t004:** Echocardiographic parameters in patients diagnosed with ST-segment elevation myocardial infarction with preserved ejection fraction (HFpEF, *n* = 100) at the time of admission and on 10th–12th day of hospital stay.

Parameter	1st Day after STEMI Onset Median (Interquartile Range)	10th–12th Day after STEMI Onset Median (Interquartile Range)	*p* Value
Left ventricular ejection fraction, %	56.0 (48.5; 61.0)	60.0 (52.0; 64.0)	0.015
Left ventricular end-diastolic volume, mL	126.0 (117.25; 142.25)	118.0 (98.0; 135.0)	0.003
Left ventricular end-systolic volume, mL	66.0 (54.0; 83.0)	62.0 (51.0; 74.0)	0.015
Left ventricular end-diastolic diameter, cm	5.5 (5.2; 5.7)	5.4 (5.23; 5.7)	0.861
Left ventricular end-systolic diameter, cm	3.9 (3.6; 4.3)	3.8 (3.5; 4.1)	0.038
Left ventricular end-diastolic volume/body surface area, mL/m2	10.88 (9.9; 11.84)	10.4 (9.44; 11.84)	0.875
Left ventricular end-systolic volume/body surface area, mL/m2	37.0 (28.0; 42.75)	32.0 (26.0; 39.0)	0.112
Pulmonary artery pressure, mmHg	25.0 (23.0; 27.5)	25.0 (23.0; 27.0)	0.086
Left ventricular end-diastolic pressure, mmHg	10.88 (9.9; 11.84)	10.4 (9.44; 11.84)	0.070
Stroke volume, mL	79.0 (70.25; 88.0)	81.0 (74.25; 90.0)	0.005
Left ventricular mass, g	241.0 (217.5; 271.0)	234.0 (213.0; 271.0)	0.141
Left ventricular mass/body surface area, g/m2	130.0 (122.0; 140.75)	124.0 (116.0; 142.0)	0.515
Left atrial diameter, cm	4.1 (3.9; 4.25)	4.1 (3.9; 4.3)	0.799
Right atrial diameter, cm	4.1 (3.9; 4.4)	4.2 (3.9; 4.4)	0.411
Right ventricular diameter, cm	1.8 (1.5; 1.9)	1.8 (1.5; 1.9)	0.855
Isovolumic relaxation time, ms	111.0 (104.0; 118.0)	110.0 (104.0; 118.0)	0.171
Early diastolic ventricular filling velocity (E, cm/sec)	57.0 (49.0; 70.0)	60.0 (47.0; 71.75)	0.662
Late diastolic ventricular filling velocity (A, cm/sec)	69.0 (59.0; 78.0)	69.5 (54.25; 78.75)	0.710
Early to late diastolic ventricular filling velocity ratio (E/A)	0.78 (0.71; 1.17)	0.79 (0.68; 1.24)	0.282
Deceleration time, ms	202.0 (170.0; 223.0)	195.0 (170.0; 221.75)	0.025
Acceleration time, ms	131.0 (114.5; 142.5)	131.0 (111.0; 137.0)	0.243
Left ventricular ejection time, ms	294.0 (279.5; 305.0)	287.0 (268.0; 300.0)	0.026
Early diastolic myocardial velocity, cm/sec	7.0 (6.0; 8.0)	6.0 (5.0; 8.0)	0.018
Late diastolic myocardial velocity, cm/sec	8.0 (6.9; 9.0)	7.95 (7.0; 9.0)	0.675
Early to late diastolic myocardial velocity ratio	0.83 (0.7; 1.14)	0.75 (0.67; 1.14)	0.009
Early mitral filling velocity to early diastolic mitral annular velocity ratio	8.59 (7.36; 10.23)	9.0 (7.67; 10.42)	0.038
Early mitral inflow velocity, cm/sec	37.0 (29.0; 45.0)	40.0 (32.0; 48.0)	0.001
Early diastolic myocardial velocity to early mitral inflow velocity ratio	1.56 (1.3; 2.0)	1.36 (1.03; 1.87)	0.001
Tei index (myocardial performance index)	0.7 (0.65; 0.77)	0.71 (0.65; 0.78)	0.758

**Table 5 diagnostics-10-00301-t005:** Serum concentrations of biomarkers in patients diagnosed with ST-segment elevation myocardial infarction with preserved ejection fraction (HFpEF, *n* = 100) and ST-segment elevation myocardial infarction with reduced ejection fraction (HFrEF, *n* = 154) at the time of admission and on 10th–12th day of hospital stay.

Parameter	Reference Range	1st Day after STEMI Onset Median (Interquartile Range)	10th–12th day after STEMI Onset Median (Interquartile Range)	*p* Value
**Patients with HFpEF**				
NT-proBNP, fmol/mL	5.0–12.0	17.84 (6.36; 60.4)	4.68 (2.5; 8.35)	0.0361
sST2, ng/mL	14.0–22.5	40.75 (26.98; 64.6)	22.19 (18.11; 25.3)	0.0001
Galectin-3, ng/mL	0.62–6.25	11.37 (9.49; 14.0)	9.05 (6.0; 10.41)	0.0001
MMP-1, ng/mL	0.91–9.34	2.14 (1.43; 5.39)	2.59 (1.72; 4.0)	0.2372
MMP-2, ng/mL	139.0–356.0	254.9 (217.2; 283.83)	295.2 (267.3; 326.65)	0.0003
MMP-3, ng/mL	2.0–36.6	7.22 (5.29; 10.6)	11.89 (8.99; 13.63)	0.0001
TIMP-1, ng/mL	11.0–743.0	899.25 (592.5; 1080.0)	853.5 (559.75; 1017.5)	0.7332
**Patients with HFrEF**				
NT-proBNP, fmol/mL	5.0–12.0	29.87 (10.65; 84.91)	18.69 (8.65; 42.76)	0.0265
sST2, ng/mL	14.0–22.5	56.45 (33.77; 75.88)	35.33 (22.79; 47.16)	0.0020
Galectin-3, ng/mL	0.62–6.25	14.59 (10.64; 16.45)	12.17 (9.12; 14.89)	0.0231
MMP-1, ng/mL	0.91–9.34	1.66 (0.73; 3.02)	2.20 (1.32; 4.51)	0.0001
MMP-2, ng/mL	139.0–356.0	289.3 (222.92; 315.31)	332.12 (297.7; 423.59)	0.0032
MMP-3, ng/mL	2.0–36.6	12.48 (10.19; 16.97)	14.67 (12.34; 17.93)	0.0035
TIMP-1, ng/mL	11.0–743.0	567.32 (412.9; 787.32)	414.22 (324.23; 668.45)	0.0039
**HFpEF vs. HFrEF (*p* Value)**				
NT-proBNP, fmol/mL	5.0–12.0	0.0056	0.0001	
sST2, ng/mL	14.0–22.5	0.0001	0.0001	
Galectin-3, ng/mL	0.62–6.25	0.0001	0.0001	
MMP-1, ng/mL	0.91–9.34	0.0075	0.82	
MMP-2, ng/mL	139.0–356.0	0.0001	0.0001	
MMP-3, ng/mL	2.0–36.6	0.0001	0.0001	
TIMP-1, ng/mL	11.0–743.0	0.0001	0.0001	

HFpEF—ST-segment elevation myocardial infarction with preserved ejection fraction, HFrEF—ST-segment elevation myocardial infarction with reduced ejection fraction, NT-proBNP—N-terminal pro-B-type natriuretic peptide, sST2—soluble suppression of tumorigenicity 2, MMP—matrix metalloproteinase, TIMP—tissue inhibitor of metalloproteinases.

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
