# Peer review of "Serum and Echocardiographic Markers May Synergistically Predict Adverse Cardiac Remodeling after ST-Segment Elevation Myocardial Infarction in Patients with Preserved Ejection Fraction"

_diagnostics, 2020, doi:10.3390/diagnostics10050301_

Round 1
Reviewer 1 Report
Dear authors,
The study is interesting as a starting point for evaluating the association between serum and echocardiographic markers in STEMI with preserved EF.
However, the main issue, with this design, is its lack of relevance. The data presented in this study is already known (changes in echocardiographic markers associated with STEMI with preserved EF, and serum markers changes). Their correlations were obvious. What would be interesting would be to evaluate whether these changes are significant compared to another group (maybe STEMI with decreased EF). Moreover, the patients previously received different treatments, and had a highly variable plethora of risk factors, making the study group heterogenous. This would have necessitated a larger study group to yield a good statistical power to the study.
Author Response
Reviewer #1
Reviewer: The study is interesting as a starting point for evaluating the association between serum and echocardiographic markers in STEMI with preserved EF.
However, the main issue, with this design, is its lack of relevance. The data presented in this study is already known (changes in echocardiographic markers associated with STEMI with preserved EF, and serum markers changes). Their correlations were obvious. What would be interesting would be to evaluate whether these changes are significant compared to another group (maybe STEMI with decreased EF).
Authors: We agree with the reviewer that comparison of patients with HFpEF (preserved ejection fraction) to those with HFrEF (reduced ejection fraction) is of high clinical relevance. Therefore, we included 154 patients with HFrEF (a sample which was recruited earlier but was not analyzed and published previously) in addition to 100 patients with HFpEF. Measurement of serum biomarkers and echocardiographic parameters was performed in the same manner in both groups.
We found that patients with HFrEF had multiple correlations of serum markers (NT-proBNP, sST2, galectin-3, MMP-3) with LVEF as well as left ventricular end-systolic volume and diameter, whereas in those with HFpEF, only NT-proBNP showed significant negative correlations with LVEF. Probably, patients with HFpEF share a number of pathophysiological pathways with those suffering from HFrEF yet cardiac remodeling in the former patient category is related to the diastolic dysfunction rather than LVEF.
In patients with HFpEF, studied biomarkers of ventricular remodeling (NT-proBNP, sST2, galectin-3, and MMP-3) do correlate with the parameters reflecting the diastolic dysfunction (myocardial stretch, transmitral flow profile, and myocardial performance (Tei) index). Contrariwise, in patients with HFrEF these markers are mainly associated with LVEF and left ventricular end-systolic volume/diameter.
Reviewer: Moreover, the patients previously received different treatments, and had a highly variable plethora of risk factors, making the study group heterogeneous. This would have necessitated a larger study group to yield a good statistical power to the study.
Authors: We agree with the reviewer and added this to the Discussion: “An important limitation of the study includes differences in risk factors and treatments between the patients resulting in a relatively heterogeneous variable; however, it was expected when considering the patient groups with distinct LVEF (HFpEF and HFrEF). A replication of our study on a larger sample would be beneficial for the field to confirm our results”.
We sincerely thank the reviewer for the valuable comments and constructive criticism.
Reviewer 2 Report
Pecherina and colleagues investigated serum biomarkers and echocardiographic parameters in patients after ST-segment elevation myocardial infarction with preserved ejection fraction. Specifically, they measured NT-proBNP, sST2, galectin-3, and MMP-1, MMP-2, MMP-3, TIMP-1 on day 1 after admission and on day 10-12 before discharge from the hospital and correlated biomarker expression with echocardiographic parameters.
They found a reduction in NT-proBNP, sST2, galectin-3, and TIMP-1 between day 1 and discharge and multiple correlations between biomarker levels and the multitude of echocardiographic parameters. Of special interest might be the negative correlation between serum NT-proBNP and LVEF detected on 10th-12th day upon STEMI onset.
The manuscript is overall well written, the study protocol well described. Some points are to consider:
- The authors frequently relate their study to development of cardiac fibrosis, but the parameters reflect more recovery after acute myocardial infarction.
- The observation interval is relatively short, which should be mentioned as a limitation.
- Although MMP levels were not outside the reference range, it should be noted and discussed that MMP-2 and -3 levels significantly increased between day 1 and 10-12, which might be a meaningful marker for cardiac remodeling in this context. Please discuss this point.
- Some small typographical errors below should be corrected:
Line 44: prognostic instead of prognostication
Line 47: released instead of releasing
Line 60: have been reported instead of are yet to be reported
Line 210: and are informative instead of and informative
Author Response
Reviewer #2
Reviewer: Pecherina and colleagues investigated serum biomarkers and echocardiographic parameters in patients after ST-segment elevation myocardial infarction with preserved ejection fraction. Specifically, they measured NT-proBNP, sST2, galectin-3, and MMP-1, MMP-2, MMP-3, TIMP-1 on day 1 after admission and on day 10-12 before discharge from the hospital and correlated biomarker expression with echocardiographic parameters.
They found a reduction in NT-proBNP, sST2, galectin-3, and TIMP-1 between day 1 and discharge and multiple correlations between biomarker levels and the multitude of echocardiographic parameters. Of special interest might be the negative correlation between serum NT-proBNP and LVEF detected on 10th-12th day upon STEMI onset.
The manuscript is overall well written, the study protocol well described. Some points are to consider:
The authors frequently relate their study to development of cardiac fibrosis, but the parameters reflect more recovery after acute myocardial infarction.
Authors: We agree with the reviewer. Indeed, in this paper we investigate acute cardiac remodeling instead of cardiac fibrosis; we therefore changed “cardiac fibrosis” to “ventricular remodeling” across the paper.
Reviewer: The observation interval is relatively short, which should be mentioned as a limitation.
Authors: We agree with the reviewer and mentioned it in the Discussion (“Another limitation is relatively short observation interval (two weeks) which should possibly be extended in replicating studies”).
Reviewer: Although MMP levels were not outside the reference range, it should be noted and discussed that MMP-2 and -3 levels significantly increased between day 1 and 10-12, which might be a meaningful marker for cardiac remodeling in this context. Please discuss this point.
Authors: We agree with the reviewer and mentioned it in the Discussion (“Intriguingly, serum levels of MMP-2 and MMP-3, although being within the reference range, significantly augmented from the admission to the discharge regardless of the ejection fraction suggesting these molecules are gradually released during ventricular remodeling. Further studies in this direction might address this point”).
Reviewer: Some small typographical errors below should be corrected:
Line 44: prognostic instead of prognostication
Line 47: released instead of releasing
Line 60: have been reported instead of are yet to be reported
Line 210: and are informative instead of and informative
Authors: We corrected the mentioned typographical errors.
We sincerely thank the reviewer for the valuable comments and constructive criticism.